# Zinc deficiency and associated factors among pregnant women's attending antenatal clinics in public health facilities of Konso Zone, Southern Ethiopia

Eskeziaw Agedew[1]*, Behailu Tsegaye[2], Agegnehu Bante[2], Eshetu Zerihun[2], Addis Aklilu[2], Meseret Girma[2], Hergewoin Kerebih[3], Mengistu Zelalem Wale[3], Mesenbet Terefe Yirsaw[3]

1 Department of Public health, College of Health Sciences, Debre Markos University, Debre Markos, Ethiopia, 2 College of Medicine and Health Sciences, Arba Minch University, Arba Minch, Ethiopia, 3 Department of Biomedical Sciences, School of Medicine, Debre Markos University, Debre Markos, Ethiopia

* esk1agid@gmail.com

## Abstract

**Data Availability Statement:** The SPSS data set was uploaded as a Supporting Information file.

### Background

Zinc is an essential mineral known to be important for the normal physiological functions of the immune system. It is one of the basic nutrients required during pregnancy for the normal development and growth of the fetus. However, Zinc deficiency during pregnancy causes irreversible effects on the newborn such as growth impairment, spontaneous abortion, congenital malformations and poor birth outcomes. Even though, the effect of Zinc deficiency is devastating during pregnancy, there is scarcity of evidence on Zinc deficiency and related factors among pregnant women in the current study area.

### Objective

To assess Zinc deficiency and associated factors among pregnant women attending antenatal clinics in public health facilities of Konso Zone, Southern Ethiopia.

### Methods

Institution based cross-sectional study was conducted among randomly selected 424 pregnant mothers. Data were collected using pre tested questionnaire (for interview part), and 5 blood sample was drawn for serum zinc level determination. Data were entered to Epi-Data version 3.1 software and exported to SPSS version 25 for analysis. Binary logistic regression analysis was computed and independent variables with a p-value $\leq 0.25$ were included in multivariable analysis. Serum zinc level was determined using atomic absorption spectroscopy by applying clean and standard procedures in the laboratory. Finally adjusted odds ratio with 95% confidence level, P-value < 0.05 was used to identify significant factors for Zinc deficiency.

**Funding:** Arba Minch University was sponsor the current study.

**Competing interests:** The authors have declared that no competing interests exist.

**Abbreviations:** ANC, Antenatal care; AOR, Adjusted odds ratio; COR, Crude odds ratio; FAAS, Flame atomic absorption spectrometer; MUAC, Mid upper arm circumference; WHO, World health organization; Zn, Zinc.

## Result

The prevalence of Zinc deficiency was found to be 128 (30.26%) with the mean serum zinc level of 0.56±0.12 g/dl. Age, 25–34 years [AOR 2.14 (1.19,3.82)], and 35–49 years [AOR 2.59 (1.15, 5.85)], type of occupation, farming [AOR 6.17 (1.36, 28.06)], lack of antenatal follow up during pregnancy [AOR 3.57 (1.05,12.14)], lack of freedom to purchase food items from market [AOR 3.61 (1.27, 10.27)], and inadequate knowledge on nutrition [AOR 3.10 (1.58, 6.08)] were factors associated with Zinc deficiency.

## Conclusion

Zinc deficiency is a public health problem among pregnant mothers in the current study area. Improving maternal nutritional knowledge, motivating to have frequent antenatal follow up, and empowering to have financial freedom to purchase food items from market were the modifiable factors to reduce Zinc deficiency. Nutritional intervention that focused on improving nutritional knowledge and insuring access to Zinc sources food items should be delivered for pregnant mothers.

## Introduction

Zinc is an essential mineral known to be important for many biological functions including protein synthesis, cellular division and nucleic acid metabolism [1]. It is crucial for the normal physiological functions of the immune system, DNA synthesis and proliferation. Furthermore, zinc is required during pregnancy for the normal development and growth of the fetus. Micronutrient deficiencies including zinc are common during pregnancy due to an increased demand of nutrients for the mother and the developing fetus [2–4]. Particularly zinc deficiency during pregnancy is associated with low birth weight and small for gestational age (SGA) [5].

Globally, it is estimated that more than 80% of pregnant women have inadequate zinc intake. World health organization (WHO) reported that the estimated prevalence of zinc deficiency ranges from 4–73% across various regions. The prevalence is low 4–7% in north America and Europe, and high in north Africa and eastern Mediterranean which, accounts 25–52%, south and central America 68% and in south and south east Asia (34–73%) [6]. Ethiopia is one of the five countries who together contribute 47% of the child deaths attributable to zinc deficiency in Africa [7].

Zinc deficiency during pregnancy causes dangerous and irreversible effects on the newborn such as growth impairment, spontaneous abortion, congenital malformations, intrauterine growth retardation (IUGR), low birth weight (LBW), preeclampsia, premature labor, prolonged labor, postpartum bleeding, delayed neurobehavioral development, delayed immune system development, and leads to increase of mortality rate [8]. In addition, it alters levels of different hormones in the circulation which are associated with the onset of labor. Zinc is essential for normal immune function; deficiency may contribute to systemic and intra-uterine infections. It also compromises infant development. Evidences showed that zinc supplementation during pregnancy reduces the risk of preterm birth especially in low income countries [9, 10].

Zinc deficiency was estimated to cause 176, 000 diarrhea deaths, 406, 000 pneumonia deaths and 207, 000 malaria deaths. The associated loss of disability-adjusted life years (DALYs) attributable to zinc deficiency amounts to more than 28 million. The burden of

disease due to Zinc deficiency is borne most heavily by countries in Africa, the Eastern Mediterranean and South-East Asia. According to a report on global and regional child mortality and burden of disease attributable to zinc deficiency, Africa suffers 58% of child deaths attributable to zinc deficiency. Ethiopia is one of the five countries contribute 47% of the child deaths attributable to zinc deficiency in Africa [11]. The burden of Zinc deficiency was higher in Ethiopia with a pooled prevalence of 59.9% [12]. Limited available reports indicated that zinc deficiency is one of a public health problem in Ethiopian among pregnant mothers. In Ethiopia, few studies were conducted to determine the prevalence and associated factors of zinc deficiency among pregnant women [13, 14]. The factors that cause zinc deficiency vary in different settings due to the variations in dietary diversity, feeding habit, socio economic difference and the burden of infectious disease like diarrhea. There is lack of information on the prevalence and contributing factors for Zinc deficiency among pregnant mothers in the current study area. Therefore, the present study was conducted to fill this gap by investigating a study among pregnant women attending antenatal care clinics at Konso Zone health facilities.

## Methodology

### Study setting

This study was carried out in Konso Zone Public health facilities, Southern Ethiopia. The live hood of the Konso community is mixed farming; crop cultivation complimented by small live stock holdings. The health care was delivered in the Konso zone in government owned health centers and district hospital [15]. The data was collected from May 1 to June, 2020.

### Study design and population

Institution based cross-sectional study was conducted among systematically selected pregnant mothers who had antenatal follow up in the public health facilities. Pregnant mothers who had known medical illness and mentally incompetent for interviews were excluded from the study.

### Sample size determination and procedures

The sample size for this study was calculated using single population proportion formula by taking the prevalence of zinc deficiency (P) = 57.4%, study done in North West Ethiopia [16], margin of error (d) = 5%, 95% confidence level and determined to be 376. By taking 15% none response rate, the final sample size became 432. From the existing public health facilities in Konso Zone three health centers were selected randomly by lottery method and the hospital was selected purposely.

The study participants were allocated proportionally to each of the selected health facilities based on the participants' flow for antenatal care. Base line data was collected from 3 heath centers and one district hospital for one month to get the actual number of pregnant mothers. Based on the collected data from the antenatal loge book, there were 242, pregnant mothers in health center one, 176 in health center two, 198 in health center three, and 374 pregnant women from the district hospital. Then each study participant was allocated proportional based on the population size. By this assumption, 105 pregnant mothers from health center one, 76 from health center two, 86 from health center three and 165 pregnant mothers were selected from the district Hospital. Finally systematic random sampling method was implemented to select the study participants for interviews and blood sample collection.

## Data collection instruments and procedures

Data was collected using pre tested structured questionnaire for survey part and blood sample was taken for serum zinc level determination. The questionnaires were prepared in English and translated in to Amharic (the national language for Ethiopia) for interviewing the participants. Women dietary diversity data was collected by using 24 hr. dietary recall methods. A well trained clinical BSc nurses and laboratory professionals were recruited to collect data from the respondents. After getting informed consent from the respondents, the actual data collection process was conducted.

## Blood sample collection

After interviewing, blood sample was collected from the pregnant mothers. Venous blood was collected using plain tube and stainless steel needles. The collected blood was allowed to clot for 20 minutes and centrifuged at 3000 revolution per minute. Visibly hemolysis blood sample was discarded. Serum was extracted and transferred immediately into screw-tub vials. Each collected serum sample was kept at -20°C until transportation. Then the stored serum was transported to Arba Minch University, Abaya campus food and chemistry laboratory.

The quality of data was assured before, during and after data collection. During data collection, supervisors and principal investigators were closely followed the day-to-day data collection process and ensure completeness and consistency of questionnaire administered each day. Blood sample quality was maintained by implementing clean and standardized procedure during laboratory analysis.

## Zinc level determination from serum sample

Serum zinc concentration was determined using flame atomic absorption spectrometry. Atomic absorption sspectroscopy was equipped with deuterium ark background correctors, hallow cathode lamps for each respective element, and air-acetylene flame. Atomic absorption spectroscopic standard solutions containing 1000 mg/L was used for preparing intermediate standard solutions (10 mg/L) by diluting in 100 ml volumetric flask using deionized water. Aseries of working standard solutions was prepared from the intermediate standard solution by 50 ml volumetric flask with 0.2, 0.4, 0.8, 1.6 mg/L concentration [17, 18]. Four points of calibration curve was established by running the prepared standard solutions in flame atomic absorption spectrometer (FAAS) Immediately after calibration, the prepared sample solution was aspirated into the FAAS instrument and direct readings of the absorbance for Zn was performed.

## Data processing and analysis

The collected data was checked visually by the investigators, then entered to Epi-Data version 3.1 software and exported into SPSS version 25 for analysis. Descriptive statistical analysis such as simple frequencies and measures of variability were used to describe the characteristics of the participants. Then the results were presented using frequencies, tables, and figures.

Wealth index was computed as a composite indicator of living standard by using Principal Component Analysis (PCA). Orthogonal rotation method using varimax was used to maximize loadings of variables on the extracted factors. Kaiser's stopping rule which considers factors with Eigen values greater than 1.0 was retained. Using the factor scores from the first principal component as weights, classified households into quintiles and calculated the mean socio-economic score for each study participant [19, 20].

Binary logistic regression analysis, COR with 95% CI, was used to see the association between each independent variable and the outcome variable. Independent variables with a p-value of ≤0.25, were included in the multivariable logistic regression analysis model. Level of statistical significance was declared at p-value < 0.05.

## Operational definitions

Zinc deficiency—Serum zinc concentration deficiency was defined as a serum zinc level of less than 56 μg/dl during the first trimester or less than 50 μg/dl during the second or third trimester [21, 22].

Dietary Diversity Score–it was measured by using Minimum Dietary Diversity–Women defined as a dichotomous indicator as low who consumed less than (≤5) food items and higher if it is greater than (≥5) out of ten defined food groups during 24 hours [23].

Wealth index–it is difficult to measure the income of pregnant mothers in study setting because the occupational status of these mothers were farmer, house wife and trader, so it is difficult to measure by using monthly income. Therefore the wealth index was measured by using principal component analysis by taking the higher score. Then each score was categorized in to five cut of points and further categorized as poor (the 1st and 2nd cut of points), medium (cut of points 3), and rich (for 4th and 5th cut of points [19, 20].

Having Nutrition knowledge: it was measured by using women nutritional knowledge assessment tool which categorized as having good if she correctly answered greater than or equal to 70% of the total knowledge assessing questions [24].

Financial freedom- of pregnant mother was measured by using tools that are used in similar setting to determine pregnant mother's financial access to purchase quality food items from the local market without their husband influence which was categorized as having freedom and not having freedom [25].

## Ethical considerations

Ethical clearance was obtained from Institutional review board (IRB) of Arba Minch University, college of Medicine and Health Sciences. The formal permission letter was obtained from the respective health facilities administrator office. Informed consent was obtained from the study participants. Needle safety procedures were in line with world health organization standard. Pregnant mothers who were zinc deficient were referred for supplemented with zinc in collaboration with respective health institutions and adequate health education was given.

## Result

### Socio demographic result

From a total of 432 participants, 424 of them were involved in this study giving a response rate of 98.15%. The mean ages of the participants were 25± 6.3years. Almost half, 217(51.2%) of the study participants were in the age range of 25–34 years. Concerning the educational status of the participants, 363(85.6%) of them had no formal education while 51, (12%) had an educational status of elementary school. Majority, 350 (82.5%) of the study participants were farmers. From the participants, 395 (93.2%) and 29 (6.8%) of them were from the rural and urban areas respectively (**Table 1**).

### Nutritional knowledge and feeding habit

Among study participants, 73.3% of them got nutritional education during antenatal care follow up and only 91(21.5%) of them had over all knowledge about balanced diets. Regarding

**Table 1. Socio-demographic characteristics of the pregnant women (n = 424) attending antenatal clinics of Konso Zone, public health facilities, Southern Ethiopia, 2020.**

| Variable | Category | Frequency(n) | Percent (%) |
|---|---|---|---|
| Age | 15–24 Years | 43 | 10.1 |
| | 25–34 Years | 217 | 51.2 |
| | Above 35 years | 164 | 38.7 |
| Mother educational level | No formal education | 363 | 85.6 |
| | Elementary school | 22 | 5.2 |
| | Secondary school | 39 | 9.2 |
| Husband educational level | No formal education | 332 | 78.3 |
| | Elementary school | 51 | 12.0 |
| | Secondary school | 41 | 9.7 |
| Occupational status | Farmer | 350 | 82.5 |
| | Government worker | 18 | 4.2 |
| | House wife | 16 | 3.8 |
| | Trader | 40 | 9.4 |
| Wealth Index | Poor | 114 | 27 |
| | Medium | 186 | 44 |
| | Rich | 123 | 29.1 |
| Residence | Rural | 395 | 93.2 |
| | Urban | 29 | 6.8 |
| Religion | Orthodox | 131 | 30.9 |
| | Protestants | 288 | 67.9 |
| | other | 4 | 1.0 |
| Ethnicity | Konso | 214 | 50.5 |
| | Derashe | 202 | 47.6 |
| | Gamo | 6 | 1.4 |
| | Amhara | 2 | 0.5 |
| Birth interval | less than 2 years | 304 | 71.7 |
| | Above 2 years | 120 | 28.3 |

the participants' knowledge about source of food, only 15.6% of them had knowledge about protein source, 141(33.3%) of them had knowledge about sources of carbohydrate and 176 (41.5%) knew about sources of vitamin.

From the study participants 297(70.0%) of them had knowledge about feeding frequency in which 104(24.5%) had a habit of 1–2 times meal frequency, 73(17.2%) of them had 3 times, and the rest 247(58.3%) of them had more than >3 times per 24 hours feeding habit (**Table 2**).

## Prevalence of Zinc deficiency and associated factors

The Prevalence of Zinc deficiency was 128(30.26%) (**Fig 1**). The mean Zinc serum level was 0.56±0.12 g/dl.

Factors including; age, type of occupation, frequency of ANC follow up, financial freedom, and nutritional knowledge were found to be statistically significant in multivariable analysis. Those participants with the age range of 25–34 years [AOR = 2.14, 95% CI (1.19–3.82)] were 2.14 times more likely have Zinc deficiency than those with bellow 25 years of old.

Participants greater than 35 years had 2.59 times [AOR = 2.59, 95% CI (1.15–5.85)] more likely to have Zinc deficiency than those with the age of below 25 years. In addition, farmer participants, those with deficient frequency of ANC follow up, those with no financial freedom

**Table 2. Nutritional knowledge and feeding habit of pregnant women (n = 424) attending antenatal clinics of Konso Zone public health facilities, Southern Ethiopia, 2020.**

| Variable | Category | Frequency | Percentage |
|---|---|---|---|
| Getting nutritional education | Yes | 311 | 73.3 |
| | No | 113 | 26.7 |
| Knowing about balanced diet | Yes | 91 | 21.5 |
| | No | 333 | 78.5 |
| Knowing about sources of protein | Yes | 66 | 15.6 |
| | No | 358 | 84.4 |
| Knowing about sources of carbohydrate | Yes | 141 | 33.3 |
| | No | 283 | 66.7 |
| Knowing about sources of vitamin | Yes | 176 | 41.5 |
| | No | 248 | 58.5 |
| Knowing feeding frequency | Yes | 297 | 70.0 |
| | No | 127 | 30.0 |
| Meal frequency | 1 to 2 times | 104 | 24.5 |
| | 3 times | 73 | 17.2 |
| | More than 3 times | 247 | 58.3 |

to purchase food items from market and those with poor nutritional knowledge, had 6.17 times [AOR = 6.17, 95% CI (1.36–28.06)], 3.57 times [AOR = 3.57, 95% CI (1.05–12.14)], 3.61 times [AOR = 3.61, 95% CI (1.27–10.27)] and 3.10 times [AOR = 3.10, 95% CI (1.58–6.08)] more likely to develop serum Zinc deficiency respectively, than their corresponding parts (Table 3). The co-variates which had crude odd ratio with p<0.25 in the binary logistic

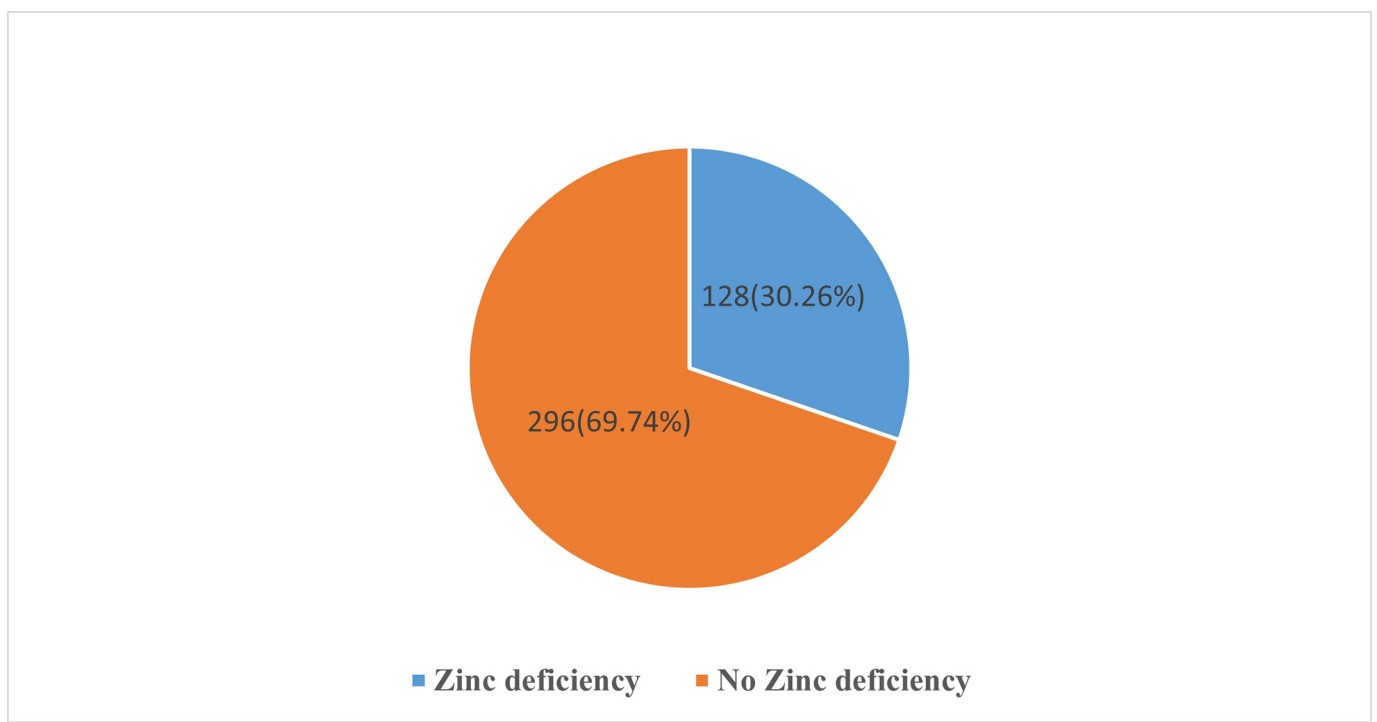

**Fig 1. Prevalence of Zinc deficiency among pregnant women (n = 424) attending antenatal care clinics of Konso Zone in public health facilities Southern Ethiopia, 2020.**

**Table 3. Zinc deficiency and associated factors among pregnant women attending antenatal clinics of Konso Zone public health facilities, Southern Ethiopia, 2020.**

| Variable | Variable category | Zinc status | | COR | | | AOR | | | P-value |
|---|---|---|---|---|---|---|---|---|---|---|
| | | Deficiency | Normal | COR | LL | UL | AOR | LL | UL | |
| **Age** | 15–24 Years | 66(51.6%) | 143(48.3%) | | 1 | | | 1 | | |
| | 25–34 Years | 48(37.5%) | 103(34.8%) | 1.52 | 0.99 | 2.32 | 2.14 | 1.19 | 3.82 | 0.011** |
| | 35–49 years | 14(10.9%) | 50(16.9%) | 4.74 | 2.52 | 8.91 | 2.59 | 1.15 | 5.85 | 0.022** |
| **Occupational status** | Farmer | 114(89.1%) | 251(85.1%) | 0.65 | 0.34 | 1.26 | 6.17 | 1.36 | 28.06 | 0.019** |
| | Trader | 4(3.1%) | 36(12.2%) | 0.37 | 0.12 | 1.18 | 2.46 | 0.77 | 7.87 | 0.129 |
| | Gov't worker | 10(7.8%) | 8(2.7%) | | 1 | | | 1 | | |
| **Residence** | Rural | 122(95.3%0 | 273(92.2%) | 1.71 | 0.68 | 4.31 | 0.59 | 0.19 | 1.82 | 0.358 |
| | Urban | 6(4.7%0 | 23(7.8%) | | 1 | | | 1 | | |
| **Wealth Index** | Poor | 34(26.6%) | 80(27.1%) | 0.63 | 0.35 | 1.14 | 0.80 | 0.39 | 1.63 | 0.536 |
| | Medium | 68(53.1%) | 118(40.0%) | 0.47 | 0.27 | 0.79 | 0.74 | 0.39 | 1.41 | 0.361 |
| | Rich | 26(20.3%) | 97(32.9%) | | 1 | | | 1 | | |
| **Frequency of ANC follow up** | Only one time | 94(73.4%) | 192(65.1%) | 0.51 | 0.22 | 1.20 | 3.57 | 1.05 | 12.14 | 0.042** |
| | From 2–3 times | 30(23.4%) | 79(26.8%) | 0.48 | 0.21 | 1.08 | 1.36 | 0.76 | 2.45 | 0.305 |
| | More than 3 times | 4(3.1%) | 24(8.1%) | | 1 | | | 1 | | |
| **MUAC** | Less than 21 CM | 20(15.6%) | 16(5.4%) | 3.23 | 1.61 | 6.46 | 0.56 | 0.26 | 1.22 | 0.147 |
| | More than 22 CM | 108(84.4%) | 279(94.6%) | | 1 | | | 1 | | |
| **Meal frequency** | 1–2 times | 28(21.9%) | 76(25.8%) | 1.24 | 0.76 | 2.03 | 1.07 | 0.60 | 1.89 | 0.89 |
| | >3 times | 100(78.1%) | 219(74.2%) | | 1 | | | 1 | 1.00 | |
| **Getting nutritional education during ANC follow up** | Yes, frequently | 88(68.8%) | 223(75.6%) | | 1 | | | 1 | 1.00 | |
| | No | 40(31.3%) | 72(24.4%) | 1.41 | 0.89 | 2.23 | 1.20 | 0.65 | 2.20 | 0.558 |
| **Dietary diversity** | Low (<5 food group) | 96(75.00%) | 235(79.7%) | 1.31 | 0.80 | 2.13 | 1.29 | 0.77 | 2.15 | 0.225 |
| | High (≥ 5 food group) | 32(25.00%) | 60(20.30%) | | 1 | | | 1 | | |
| **Having financial freedom** | Yes | 114(89.10) | 264(89.5%) | | 1 | | | 1 | | |
| | No | 14(10.9%) | 31(10.5%) | 1.05 | 0.54 | 2.04 | 3.61 | 1.27 | 10.27 | 0.016** |
| **Having Nutrition knowledge** | Poor | 82(64.1%) | 172(58.3%) | 1.28 | 0.83 | 1.96 | 3.10 | 1.58 | 6.08 | 0.001** |
| | Good | 46(35.9%) | 123(41.7%) | | 1 | | | 1 | | |

Key:

** Significant factors, 1-referance variable category, MUAC = mid upper arm circumference

regression were pregnant mother educational status, marital status, gestational age, pregnant mothers parity, and birth interval of the mothers.

## Discussion

The prevalence of Zinc deficiency was found to be 30.26%. This figure shows that, one third of pregnant women were suffer with zinc deficiency which could be considered as a public health micronutrient problem in the study area based International Zinc Nutrition Consultative Group (IZiNCG) considered as a public health concern when the prevalence of low serum zinc concentrations is greater than 20% [26].

The prevalence of Zinc deficiency in the current study (30.26%) was lower than a study done in Gondar, North West Ethiopia [16] and in Sidama, Southern Ethiopia [27], which accounts 57.4% and 53.0% the burden of Zinc deficiency higher in Ethiopia with pooled prevalence 59.9% [12] of participated pregnant women had zinc deficient respectively. It was also much lower than study finding from other countries, including; Kenya 66.9% [28] Cameroon 82% [29] and India 73.5% [30].

The lower prevalence of zinc deficiency (30.26%) in the current study setting is due to the availability diversified food items which have good sources of zinc. In addition animal source

food items consumption habit of community higher in the current setting but in other part of the country there was dominant consumption habit of cereal grain based food with higher level of fiber which reduce zinc absorption.

As the recent evidence indicated that dietary habit of the community in North West Ethiopia [16] was dominant 99.7% dependency on consumption of cereals. This leads to poor absorption of Zinc due high content of phytates [31]. In addition, there was difference in study area of residence as urban-rural difference as study done in Southern Ethiopia and India focused on rural pregnant mothers due this the prevalence may be overestimated in previous studies [27, 30].

The prevalence of Zinc deficiency in the present study was almost relatively comparable with studies conducted in Sudan (38%) [32] and Vietnam (29%) [33]. However, it is higher than result from Bangladesh (14.7%) [34]. This difference probably is due to cultural differences in food preparation and feeding habit. Further, the study may underestimated the problem as they only included pregnant women in early pregnancy in the study at Bangladesh [16, 34].

According to this study some associated factors with Zinc deficiency were modifiable factors including; having poor nutrition knowledge [AOR 3.10 (1.58, 6.08)], lack of antenatal follow up during pregnancy [AOR 3.57 (1.05, 12.14)]. Pregnant mothers who had no frequent antenatal follow up did not get education and counselling related to nutrition at the health facilities this leads to poor practice of feeding habit of Zinc sources food items to get optimal zinc intake from their daily diet consumption [35]. Pregnant women who were in advanced age (35–49 years) were 2.9 times [AOR 2.59 (1.15, 5.85)] more likely develop Zinc deficiency as compared with those whose were in young age group. This may be due to the fact that serum zinc level reaches peak during adolescence and young adulthood, and then declines as the age of individual increase [35].

Pregnant mothers who were farmer in their occupation were 6.17 times [AOR 6.17 (1.36, 28.06)] more likely develop Zinc deficiency as compared those who were government workers. This is due the fact that pregnant women who engaged in farming activities were lived in rural areas and this expose them to had low nutritional awareness, and thus will be more vulnerable to food shortage and micronutrient deficiencies. Further, pregnant women from rural areas were more likely to be involved in laborious activities, as chronic overexertion is a predisposing factor to maternal nutritional depletion, leads to Zinc deficiency [27]. Participants who had no financial freedom to purchase food items from market were 3.61 times [AOR 3.61 (1.27, 10.27)] more likely develop Zinc deficiency as compared to those who had financial freedom. This might be due to as women did not get access to finance for purchasing variety of food items from market to feed diversified food item which were rich in zinc.

## Limitations of the study

Due to cross sectional nature of study design it is difficult to establish causal association between factors and outcomes.

## Conclusion

The prevalence of Zinc deficiency was a significant public health problem for pregnant mothers in current study area. Improving maternal nutritional knowledge, motivating to have frequent antenatal follow up, and empowering to have financial freedom to purchase food items from market were the modifiable factors to reduce Zinc deficiency. Giving special emphasis for mothers who are farmers and advanced age groups is important for reducing Zinc

deficiency. Nutritional intervention that focused on improving nutritional knowledge and access to Zinc sources food items should be delivered for pregnant mothers.

## Supporting information

**S1 Table.**
(SAV)

## Acknowledgments

First, we would like to acknowledge all the study participants for their willingness. We extend our thanks to the staffs of Konso Zone public health facilities for their cooperativeness to provide all the necessary baseline information, which, were important for this study.

## Author Contributions

**Conceptualization:** Eskeziaw Agedew, Eshetu Zerihun, Addis Aklilu, Hergewoin Kerebih, Mesenbet Terefe Yirsaw.

**Data curation:** Eskeziaw Agedew, Behailu Tsegaye, Agegnehu Bante, Eshetu Zerihun, Meseret Girma, Hergewoin Kerebih.

**Formal analysis:** Eskeziaw Agedew, Addis Aklilu.

**Methodology:** Eskeziaw Agedew, Behailu Tsegaye, Agegnehu Bante, Addis Aklilu, Mengistu Zelalem Wale.

**Project administration:** Eskeziaw Agedew, Behailu Tsegaye, Meseret Girma.

**Validation:** Mengistu Zelalem Wale.

**Writing – original draft:** Eskeziaw Agedew, Eshetu Zerihun, Meseret Girma, Mesenbet Terefe Yirsaw.

**Writing – review & editing:** Eskeziaw Agedew, Behailu Tsegaye, Agegnehu Bante, Eshetu Zerihun, Addis Aklilu, Meseret Girma, Hergewoin Kerebih, Mengistu Zelalem Wale, Mesenbet Terefe Yirsaw.

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
