## [Decision Letter · Decision Letter 0]

10 Dec 2021

PONE-D-21-34642Zinc deficiency and associated factors among pregnant women’s attending antenatal clinics in public health facilities of Konso, Southern EthiopiaPLOS ONE

Dear Dr. Getahun,

Thank you for submitting your manuscript to PLOS ONE. After careful consideration, we feel that it has merit but does not fully meet PLOS ONE’s publication criteria as it currently stands. Therefore, we invite you to submit a revised version of the manuscript that addresses the points raised during the review process.

We look forward to receiving your revised manuscript.

Kind regards,

Ammal Mokhtar Metwally, Ph.D (MD)

Academic Editor

PLOS ONE

Journal Requirements:

● A clean copy of the edited manuscript (uploaded as the new *manuscript* file).

[First, all authors would like to acknowledge Arba Minch University for funding of the study. Secondly, we would like to acknowledge all the study participants for their willingness. We extend our thanks to the staffs of Konso public health facilities for their cooperativeness to provide all the necessary baseline information, which, were important for this study.]

 [The funders had no role in study design, data collection and analysis, decision to publish, or preparation of the manuscript.]

Reviewers' comments:

Reviewer's Responses to Questions

**Comments to the Author**

1. Is the manuscript technically sound, and do the data support the conclusions?

Reviewer #1: Yes

Reviewer #2: Partly

2. Has the statistical analysis been performed appropriately and rigorously? 

Reviewer #1: Yes

Reviewer #2: I Don't Know

3. Have the authors made all data underlying the findings in their manuscript fully available?

Reviewer #1: Yes

Reviewer #2: No

4. Is the manuscript presented in an intelligible fashion and written in standard English?

Reviewer #1: Yes

Reviewer #2: No

5. Review Comments to the Author

Reviewer #1: This is an interesting study and the authors have collected a unique dataset using cutting edge methodology. This is a clear, concise, and well-written manuscript. The paper is generally well written and structured

Reviewer #2: PONE-D-21-34642

Zinc deficiency and associated factors among pregnant women’s attending antenatal clinics in public health facilities of Konso, Southern Ethiopia

PLOS ONE

*Comments to the Author

1. Is the manuscript technically sound, and do the data support the conclusions?

Reviewer response to question #1:

The manuscript deals with the important issue of zinc deficiency in pregnant women. The methodology utilized to calculate prevalence ideally should be based on a random sample of the study population. In 2015, an Ethiopia National Micronutrient Survey was implemented with serum samples from over 3,300 subjects, with a 75% overall prevalence of zinc deficiency. A further systematic review of Ethiopian studies on zinc deficiency was published in 2019, reporting a 60% pooled prevalence of zinc deficiency in pregnant women. Neither of these references are cited in the manuscript. The authors should discuss more thoroughly the possible reasons why they found only 30% prevalence of zinc deficiency in their sample of pregnant women.

In this manuscript, there are several methodological issues that should be addressed by the authors.

(1) Sampling of study sites. Institution-based sampling could have introduced bias, since a representative random sample of the population was not used. The sample was selected from among women attending prenatal care at one hospital or one of three health centers, which were randomly selected from a group of one hospital and nine health centers. The authors should state what proportion of the total population of pregnant women attended formal prenatal care services. They should clearly describe what group of pregnancy women is represented by the study, and what, if any, possible biases were introduced by sampling from an institutional population.

(2) Sampling of study subjects. The methods for sampling pregnant women within study sites is not described; this should be discussed in detail in the manuscript. The period of time during pregnancy during which blood samples are taken is an important factor for blood assays; this should be described in the manuscript. The authors did take into account timing during pregnancy by assigning different cut-off points of zinc deficiency by first trimester versus second or third trimester of pregnancy. A bibliographic reference needs to be cited to support the cut-off levels chosen. The variation in timing of taking blood samples and interviewing subjects should be discussed in the manuscript, possibly in the limitations section.

(3) Sample size. The sample size appears to have been calculated correctly to estimate prevalence. The number of study subjects selected at each site was proportional to the size of the health facility. Thus, it is possible that there was oversampling from the hospital. The authors should discuss the similarity or differences between the characteristics of pregnant women who attended prenatal care in a hospital versus a health center.

(4) I am unable to comment on the laboratory procedures for the zinc assay.

(5) Co-variates. The important independent variables on nutrition knowledge should be be more well-defined in the manuscript, such as “getting nutritional education”, “knowing about balance diet,” knowing about food sources of various types of nutrients, and “meal frequency,” and “dietary diversity.” For example, if a standard 24-hour recall on dietary intake was conducted to calculate the WHO indicator on “dietary diversity,” it should be reported with the appropriate reference cited. Other independent variables such as “wealth index” and “financial freedom” also should be described as to how they were calculated.

(6) The personal interviews of study subjects in this research should have included a 24-hour dietary recall, which would have identified any consumption of zinc-rich foods. Also, the interview questionnaire of pregnant women should have inquired about specific knowledge of what foods are rich in zinc. If this was done, it should be reported in the manuscript.

(7) Conclusions. The authors need to state the conclusions more precisely and should avoid a tendency for phrases that suggest causality, given that this observational cross-sectional study is only able to identify “associations” between independent variables and the outcome variable of zinc deficiency.

*2. Has the statistical analysis been performed appropriately and rigorously?

Reviewer response to question #2

(1) Data analysis methods need to be described more clearly. The acronym COR is not spelled out.

(2) The format of presentation is not clear on Table 3 provided. The acronym AOR should be spelled out. The co-variates with an association of p<.25 which were used for adjustment in the logistic regression should be listed in the text or at the bottom of the table. The final best-fitting logistic model should be clearly shown.

(3). Discussion of the findings should be reviewed to align with an understanding of the statistical methods used, such as the statement: “The current study showed that participants whose age was advanced, 35-49 years [AOR 2.59 (1.15, 5.85)] were 2.59 times more likely develop Zink deficiency as compared with those whose age was below 35 years.” This statement should be modified to state that the AOR of 2.59 of the older age group is in comparison to the youngest age group which was used as the reference group.

*3. Have the authors made all data underlying the findings in their manuscript fully available?

*4. Is the manuscript presented in an intelligible fashion and written in standard English?

Reviewer response to question #4

The manuscript needs a significant review and correction of English grammar, spelling, and punctuation. There is a repeat of one sentence with the phrase “… contributes 47% of child deaths."

6. PLOS authors have the option to publish the peer review history of their article (what does this mean?). If published, this will include your full peer review and any attached files.

Reviewer #1: **Yes: **MAHMOUD AL-MASAEED

Reviewer #2: **Yes: **Dr. Laura C. Altobelli, DrPH, MPH

---

## [Author Response · Author response to Decision Letter 0]

8 Feb 2022

Dear Reviewers , we address your valued comments and feed back in revised manuscript.

---

## [Decision Letter · Decision Letter 1]

22 Jun 2022

Zinc Deficiency and Associated Factors among Pregnant Women’s Attending Antenatal Clinics in Public Health Facilities of Konso Zone, Southern Ethiopia

PONE-D-21-34642R1

Dear Dr. Getahun,

We’re pleased to inform you that your manuscript has been judged scientifically suitable for publication and will be formally accepted for publication once it meets all outstanding technical requirements.

Kind regards,

Ammal Mokhtar Metwally, Ph.D (MD)

Academic Editor

PLOS ONE

Additional Editor Comments (optional):

Reviewers' comments:

Reviewer's Responses to Questions

**Comments to the Author**

1. If the authors have adequately addressed your comments raised in a previous round of review and you feel that this manuscript is now acceptable for publication, you may indicate that here to bypass the “Comments to the Author” section, enter your conflict of interest statement in the “Confidential to Editor” section, and submit your "Accept" recommendation.

Reviewer #2: All comments have been addressed

Reviewer #3: All comments have been addressed

2. Is the manuscript technically sound, and do the data support the conclusions?

Reviewer #2: Yes

Reviewer #3: Yes

3. Has the statistical analysis been performed appropriately and rigorously? 

Reviewer #2: Yes

Reviewer #3: Yes

4. Have the authors made all data underlying the findings in their manuscript fully available?

Reviewer #2: Yes

Reviewer #3: Yes

5. Is the manuscript presented in an intelligible fashion and written in standard English?

Reviewer #2: No

Reviewer #3: Yes

6. Review Comments to the Author

Reviewer #2: (No Response)

Reviewer #3: I congratulate authors for their valuable work. Research objective was clearly explained. Reviewer comment were well address.

7. PLOS authors have the option to publish the peer review history of their article (what does this mean?). If published, this will include your full peer review and any attached files.

Reviewer #2: **Yes: **Laura Altobelli

Reviewer #3: No

---

## [Editor Report · Acceptance letter]

27 Jun 2022

PONE-D-21-34642R1 

Zinc Deficiency and Associated Factors among Pregnant Women’s Attending Antenatal Clinics in Public Health Facilities of Konso Zone, Southern Ethiopia 

Dear Dr. Getahun:

I'm pleased to inform you that your manuscript has been deemed suitable for publication in PLOS ONE. Congratulations! Your manuscript is now with our production department. 

Kind regards, 

on behalf of

Professor Ammal Mokhtar Metwally 

Academic Editor

PLOS ONE